

# Addressing cyberbullying in Urdu tweets: a comprehensive dataset and detection system

Farah Adeeba[1], Muhammad Irfan Yousuf[1], Izza Anwer[2], Sardar Umair Tariq[1], Abdullah Ashfaq[1] and Malik Naqeeb[1]

[1] Department of Computer Science, University of Engineering and Technology Lahore, Lahore, Punjab, Pakistan
[2] Department of Transportation Engineering and Management, University of Engineering and Technology Lahore, Lahore, Punjab, Pakistan

## ABSTRACT

The prevalence of cyberbullying has reached an alarming rate, affecting approximately 54% of teenagers who experience various forms of cyberbullying, including offensive hate speech, threats, and racism. This research introduces a comprehensive dataset and system for cyberbullying detection in Urdu tweets, leveraging a spectrum of machine learning approaches including traditional models and advanced deep learning techniques. The objectives of this study are threefold. Firstly, a dataset consisting of 12,500 annotated tweets in Urdu is created, and it is made publicly available to the research community. Secondly, annotation guidelines for Urdu text with appropriate labels for cyberbullying detection are developed. Finally, a series of experiments is conducted to assess the performance of machine learning and deep learning techniques in detecting cyberbullying. The results indicate that fastText deep learning models outperform other models in cyberbullying detection. This study demonstrates its efficacy in effectively detecting and classifying cyberbullying incidents in Urdu tweets, contributing to the broader effort of creating a safer digital environment.

## INTRODUCTION

Twitter being used by approximately 42% of people worldwide (*Enough is Enough, 2022*) proves that social media is an integral part of this dynamic world. Unfortunately, along with social media's rising popularity, negative practices particularly cyberbullying have also increased significantly. Since the start of the Covid-19 lockdown, there has been a staggering 70% rise in the amount of bullying and hate speech among teens and children (*Sampathkumar & Shwayder, 2020*). Cyberbullying is a social problem that involves various forms of harassment, threats, intimidation, sexting, racism, and offensive aggressiveness that uses electronic means (*Abaido, 2020*).

Cyberbullying is a global issue, and it affects individuals in various countries, including Pakistan. The Prevention of Electronic Crimes Act (PECA) (*Prevention Electronic Crimes Act, 2016*) of Pakistan guarantees strict punishments for online bullies. Due to volume of

Corresponding author
Farah Adeeba,
farah.adeeba@uet.edu.pk

daily tweets on Twitter, it is cumbersome task to manually detect cyberbullying. Various cyberbullying detection systems have been developed for different languages, however, there are very few systems that can work for the Urdu language, despite the following facts: (a) national language & lingua franca of Pakistan, (b) 40% of female Internet user of Pakistan experienced cyberbullying (*Haider, 2020*). Therefore, there is dire need to conduct research on cyberbullying detection for Urdu language to develop a cyberbullying system that can identify and classify cyberbullying in Urdu tweets.

The primary objectives of this study are listed below:

1. To design and develop a comprehensive cyberbullying detection *corpus* in the Urdu language, encompassing various types or categories of cyberbullying behaviors.
2. To develop data annotation guidelines that enable the distinction between different types of cyberbullying and non-cyberbullying content.
3. To assess thoroughly the effectiveness of both machine learning techniques and state-of-the-art deep learning techniques, augmented by various features, in detecting cyberbullying on Twitter.

To fulfill these objectives, this study has made following contributions:

1. Cyberbullying *corpus*: A finely annotated cyberbullying detection *corpus* for the Urdu language has been developed, comprising 12,759 tweets labeled as "insult," "offensive," "profane," "name-calling," "threat," "curse," or "none." The publicly available *corpus* (*Adeeba, 2024*) serves as a valuable resource to initiate research on detecting bullying content in the Urdu language, addressing the existing research gap.
2. Generic data annotation guidelines: A systematic iterative approach has been employed to develop guidelines for each class, enabling the disambiguation of uncertain tweets. The language-independent guidelines presented in this study can not only enhance the developed *corpus* but also assist in creating new datasets for Urdu and other languages.
3. Assessment of machine learning techniques for cyberbullying detection: Experimental evaluations have been conducted using various traditional techniques as well as state-of-the-art deep learning techniques, including LSTM (Long Short Term Memory) and fastText. The aim is to identify the most effective techniques for cyberbullying detection.

These contributions significantly advance the field by providing a dedicated cyberbullying *corpus* for Urdu. The rest of the article is organized as follows: "Related Work" provides an overview of the existing studies. "Cyberbullying Detection Corpus for Urdu" presents the development of cyberbullying *corpus* and specifications. "Experimentation" describes the details of experimentation including preprocessing, feature selection and classification techniques. "Results and Discussion" discusses the results and error analysis whereas "Conclusion" concludes the research work.

## RELATED WORK

Numerous studies have explored the identification of inappropriate or contentious material across social media platforms. These investigations encompass a range of areas such as identifying sentiment analysis, hate speech, offensive language, and addressing issues like cyberbullying. According to *Hosseinmardi et al. (2015)* types of social bullying occur on social media include flaming (profanity, offensive), name-calling and threat. A significant amount of research on cyberbullying has primarily focused on analyzing English-language content. However, there have also been numerous studies examining cyberbullying detection in both Euorpean and Asian languages. Given the extensive body of research in this area, there is a growing need for a comprehensive survey that thoroughly assesses the existing research. So, research related to cyberbullying detection in Urdu language is only presented in this section.

An overview of cyberbullying detection studies conducted on Urdu language is presented in Table 1. Research studies are ordered by the scripts of dataset used for detection *i.e.*, Roman Urdu (written using English alphabets) and Urdu (written in Arabic script).

Following notable observations can be made from the Table 1:

The studies encompass both the Arabic script and the Latin (Roman Urdu) script. It's worth mentioning that the Roman Urdu employs English alphabets to represent Urdu text, While Arabic script employs Arabic letter. This highlights the diverse approaches and linguistic variations considered in the research.

Majority of datasets created for cyberbullying detection are relatively small, specifically for Urdu written in Arabic script *i.e.*, consisting of around 7,000 tweets. This highlights the importance of developing a comprehensive dataset for the Urdu written in Arabic script, which serves as the primary focus of the current study. The cyberbullying data is annotated with multiple categories to detect specific type of cyberbullying.

Table 1 reveals a limited number of studies focusing on cyberbullying detection for Urdu language. This indicates that the research community has allocated little attention to this specific natural language processing (NLP) task. Furthermore, all of these studies have been undertaking within the past three years, specifically during 2020–2023. This suggests that the interest in detecting cyberbullying content in the Urdu language has emerged relatively recently.

An overview of the techniques and features utilized in the conducted studies is shown in Table 2. A notable observation from the table is that a significant portion of the research predominantly employed classical machine learning techniques, while the utilization of deep learning techniques is relatively less prevalent. Recognizing the effectiveness of long short-term memory (LSTM) and fastText in achieving superior performance, as evidenced by prior research, our study strategically incorporates these advanced techniques. LSTM excels in capturing temporal dependencies, crucial for understanding the nuanced dynamics of cyberbullying in social media conversations. Simultaneously, fastText's proficiency in representing subword units enhances its capability to decipher the intricacies of languages like Urdu, commonly found in colloquial expressions.

**Table 1  Summary of the existing research for cyberbullying detection in the Urdu language.**

| Study | Script | Tweets | Cyberbullying categories |
|---|---|---|---|
| *Talpur & O'Sullivan (2020)* | Roman Urdu | 17,000 | Cyberbullying, none |
| *Dewani, Memon & Bhatti (2021a)* | Roman Urdu | 3,000 | Threat, racism, insult, sexual talk, curse, defamation, personality characteristics, defamation |
| *Rasheed, Anwar & Khan (2022)* | Roman Urdu | 5,000 | Cyberbullying, non-cyberbullying |
| *Raza, Khan & Soomro (2021)* | Urdu | 2,400 | Abusive, non-abusive |
| *Amjad et al. (2021)* | Urdu | 3,564 | Threatening, non-threatening |
| *Khan & Qureshi (2022)* | Urdu | 7,625 | Non-offensive, offensive, body shaming/racial abuse, political abuse |
| *Mehmood et al. (2022)* | Urdu | 3,564 | Threatening, non-threatening |
| *Amjad et al. (2022)* | Urdu | 3,500 | Abusive, non-abusive |

**Table 2  Overview of existing classification techniques and feature utilization for the detection of cyberbullying.**

| Study | Classical techniques | Deep learning techniques | Features |
|---|---|---|---|
| *Talpur et al. (2020)* | NB, LR, KNN, DT | — | BOW, n-gram, Word2Vec |
| *Dewani, Memon & Bhatti (2021b)* | — | LSTM, CNN | Word embeddings |
| *Amjad et al. (2021)* | SVM, LR, RF, Ada-Boost | fastText, LSTM, CNN | N-grams, TF-IDF |
| *Raza, Khan & Soomro (2021)* | LR, DT, ANN | — | TF-IDF |
| *Khan & Qureshi (2022)* | NB, LR, KNN, DT | — | TF-IDF, BOW |
| *Rasheed, Anwar & Khan (2022)* | SVM, NB, LR, KNN | — | TF-IDF |
| *Mehmood et al. (2022)* | SVM, LR, BNB, Stacking | LSTM, CNN | TF-IDF, BOW |
| *Amjad et al. (2022)* | SVM, LR, RF, Ada-Boost | fastText, LSTM, CNN | N-grams, TF-IDF |
| *Dewani et al. (2023)* | SVM, NB, LR, DT, AdaBoost, Bagging | — | TF-IDF, Unigram, bigram, trigram |

**Note:**
NB, Naïve Bayes; BNB, Bernoulli Naïve Bayes; SVM, support vector machine; LR, logistic regression; DT, decision tree; KNN, K-nearest neighbors; CNN, convolutional neural network; RF, random forest; LSTM, long short term memory; BOW, bag of words; TF-IDF, term frequency inverse document frequency.

It is evident from the literature review that there is scarcity of studies focusing on cyberbullying detection in Urdu highlights the need for further research and attention in this area. Additionally, the utilization of classical machine learning techniques is prevalent, while the exploration of deep learning techniques for cyberbullying detection remains limited. Therefore, there is a clear opportunity to evaluate the effectiveness of deep learning approaches in addressing this pressing issue. By advancing the understanding and development of effective cyberbullying detection methods in the Urdu language, we can contribute to creating a safer and more inclusive digital environment for Urdu-speaking individuals.

## CYBERBULLYING DETECTION *CORPUS* FOR URDU

This section provides a concise overview of the development process for the cyberbullying detection *corpus*, which encompasses three main steps: (1) data scraping, (2) annotation guidelines development, and (3) data annotation. The following subsections outline the details of each step in the *corpus* development process.

## Data scraping

The initial step in developing the research *corpus* involved scraping Urdu tweets, specifically focusing on tweets originating from Pakistan. For this purpose, the Twitter-Scraper module from the SNS library in Python was utilized.

The data collection process was carried out at the district level, focusing on the target region of Pakistan. To identify the specific districts, geo-coordinate locations were extracted from reliable sources such as Google Answers and *LatLong.net (2022)*. The districts were ranked based on population to ensure a representative sample encompassing various regions. The list of all districts was obtained from the official website of the Pakistan Bureau of Statistics (*PBS, 2020*). A maximum of 6,000 tweets were collected for each district, aiming to gather a substantial volume of data for analysis and ensuring diverse representation from different regions within Pakistan.

To capture a wide range of tweets and observe potential variations over time, the data collection timeframe spanned from January 2022 to July 2022. This extended duration allowed for the collection of a significant number of tweets and facilitated the analysis of temporal patterns and content changes. To ensure an even distribution of tweets throughout the collection period, approximately 1,000 tweets were targeted for each month. This systematic approach ensured the acquisition of a balanced dataset, covering different months within the specified timeframe.

To enhance the dataset's coverage and address potential imbalances, tweets containing cyberbullying terms were selectively included for data augmentation. Careful consideration was given to choosing terms that would capture a broad range of content relevant to the research focus. By incorporating tweets specifically mentioning these terms, the coverage and representation of the dataset were improved, enabling a more comprehensive analysis. Furthermore, this approach helped balance the dataset by ensuring adequate representation of various topics and aspects, thereby mitigating potential bias stemming from an uneven distribution of tweets across different themes or subjects.

In conjunction with data augmentation, the training dataset underwent a process of shuffling. The order of tweets was randomized to introduce variability into the model's exposure during each training epoch. This strategy aimed to prevent the model from learning patterns associated with the order of the data and, consequently, reduced the risk of overfitting.

By following the outlined data collection steps, a diverse and representative dataset encompassing tweets from various districts in Pakistan was acquired. The inclusion of the timeframe, shuffling of tweets, and data augmentation (fetching tweets against cyberbullying specified terms) ensured the incorporation of a wide range of tweets, ultimately enhancing the overall quality of the dataset.

## Guidelines development

Annotation guidelines development process was an iterative cycle that involves repeating the steps to achieve a high-quality annotated dataset. The complete cycle includes defining annotation guidelines, annotating tweets, conducting completeness and correctness

checks, inter-annotation comparison for diversity and cross-check, reviewing conflicting cases, refining rules, and repeating the process as necessary. This iterative approach helped to improve the accuracy and reliability of the annotated data with each iteration.

The annotation guidelines were developed through an iterative process. Initially, a random sample of 10% from the collected tweets was selected. Three native Urdu annotators were then assigned the task of independently labeling each tweet as "insult," "offensive," "name-calling," "threat," "curse," "profane," or "None." Simultaneously, the annotators were instructed to create an initial set of guidelines based on their individual annotations.

The annotated results were compared and discrepancies in the labeled data sample were discussed among the annotators. This collaborative discussion helped refine and finalize the guidelines. Through this iterative process, a comprehensive set of guidelines for annotation was successfully developed. Guidelines developed for data annotation are presented in Table 3.

## Data annotation

The data annotation process is pivotal in generating a dataset of utmost quality. It encompasses a systematic procedure for meticulously assigning precise labels to each tweet within the *corpus*, ensuring the creation of a dataset of high quality and accuracy. The annotation process ensures that the dataset is appropriately labeled to facilitate subsequent analysis and model training. The annotation process was conducted with three native Urdu annotators. As a result of this process tweets were annotated with label of "insult", "curse", "name-calling", "profane", "offensive", "threat" or "none".

The following section provides further details regarding the annotation process employed in this research.

Tweets annotation: Once the annotation guidelines were established, the tweets were annotated accordingly. Each tweet was carefully reviewed and labeled based on its content and relevance to the research topic. The annotation process involves assigning appropriate label to each tweet.

Complete and correctness check: After annotating the tweets, a completeness and correctness check was performed to check the quality of annotated data. This step involves reviewing the annotated dataset to identify any tweet with missing label or incorrect label (label other than defined in guidelines). It was crucial to address any errors or discrepancies at this stage to maintain the quality of the annotated dataset.

Review conflicts: Similar to the initial level, conflicts arising from the annotators' decisions were identified, and consensus was reached through discussion. In cases where conflicts persisted on the labeling of a tweet during the annotation process, a third annotator was brought in for discussion. The annotators collectively deliberated on the differing opinions and arrived at a consensus to assign the appropriate label based on a majority vote. This approach ensured that the final label for each tweet accurately reflected the agreed-upon decision among the annotators.

**Table 3 Guidelines for cyberbullying annotation.**

Guidelines for insult

1. Insulting an individual/group/entity
2. Degrading individual/group/entity
3. Consider the local context when annotating implicit insults

Guidelines for name-calling

1. Calling someone with disrespectful names
2. Used derogatory language toward others

    a. Racial slur(s) content is used
    b. Religious insulting words like Heathen, Infidel, Kafir, Bible-thumper, Mullah are used
    c. Ableist slur(s) is used *e.g.*, Retard, Cripple, Psycho, Lame, Spaz.
    d. Size-related insult of a person
    e. Age-based insult of a person

Guidelines for offensive

1. Offensive language directed towards an individual, group, or entity
2. A decision written using offensive terms.
3. Blaming an individual/entity/group for an act
4. Consider the local context to determine that tweet can deeply hurt, upset, or incite hate and anger among individuals.

Guidelines for profane

1. Use profane words
2. Contain sexual content
3. Use vulgar terms

Guidelines for curse

1. Contain curses
2. Express negative supernatural prayers
3. Express negative prayer

Guidelines for threat

1. Violence against individual or group
2. Aggression towards individual or group
3. Explicit threat against someone well-being
4. Encouraging or inciting individuals to engage in violence against others
5. Contains content related to attempting to blackmail or coerce an entity to perform certain tasks against their will
6. Consider the local context when deciding whether a tweet contains implicit threat or not

(Continued)

Guidelines for none

1. Reflects or expresses a positive attitude
2. Express positive feeling,
3. Express a positive thought
4. Express positive comment
5. Express positive admiration towards an entity without any negation or sarcasm.
6. Express love
7. Express praise
8. Express well-being
9. Express positive prayer
10. Reflects sadness
11. Express anger
12. Express hate
13. Conversational statement
14. If a tweet or content expresses both neutral and positive sentiments, it is still labeled as positive.
15. A tweet should be labeled as None if it does not reflect any bullying aspect and is either positive or neutral sentiment

**Table 4** Distribution of tweets.

| Category | Tweets |
|---|---|
| Insult | 1,532 |
| Offensive | 995 |
| Name-calling | 738 |
| Profane | 397 |
| Threat | 92 |
| Curse | 76 |
| None | 8,598 |
| **Total** | **12,428** |

## Specification of developed *Corpus*

The specification of developed *corpus* is presented in Table 4. Around 12,500 Urdu tweets were finalized with different type of cyberbullying for cyberbullying detection system. The majority of tweets fall into the 'None' category (8,598), followed by 'Insult' (1,532), 'Offensive' (995), 'Name-calling' (738), 'Profane' (397), 'Threat' (92), and 'Curse' (76). The total number of tweets in the dataset is 12,428.

## Example tweets from few categories

Table 5 presents a collection of annotated tweets written in Urdu, along with their corresponding Roman equivalents and English translations. This subset specifically showcases tweets annotated against categories such as 'Name-calling' and 'Threat'. Examining these examples provides a nuanced exploration of the labeled data, offering

**Table 5 Example tweets from developed *corpus*.**

**Name-calling**

| 1 | Nastaliq Urdu | دو کوڑی کے ڈنگر عمران خان کو مناظرے کا چیلنج کر رہے ہیں #امپورٹڈ_حکومتِ_نامنظور |
| | RU Urdu | Doo kori kay dangar Imran khan ko manazrey ka challenge ker rehey heen. Imported hakoomat namanzoor |
| | English translation | Two bit as*es are challenging Imran Khan for a debate |
| 2 | Nastaliq Urdu | بھائی موٹے مبین کی طرف سے یوتھیوں سے اظہار تعزیت (ﷺﷺ) |
| | RU Urdu | Bhai motey Mubeen ki taraf sey youthyon sey izhar taziyat |
| | English translation | Condolences from Fat Mubeen to youthia |

**Threat**

| 1 | Nastaliq Urdu | میں تمہیں نسل کے ساتھ مٹانے کے لئے آنے والا ہوں |
| | RU | Mein tumeen nasal ky saath mataney ky liyey aney wala hon |
| | English translation | I am coming to wipe you out with the generation |
| 2 | Nastaliq Urdu | اب تک میں تمہیں چھوڑ رہا تھا، لیکن اب میں تمہیں قتل کر دوں گا |
| | RU | Ub tak mein tumeen choor reha tha, lakn ub mein tumeen qatal kr don ga |
| | English translation | Until now I was leaving you, but now I will kill you |

**Curse**

| 1 | Nastaliq Urdu | ملک کی بدنامی کرنے والے کو اللہ ننگی موت دے۔ |
| | RU | Mulk ki badnami kerney waley ko Allah nangi moot dey |
| | English translation | May Allah give a disgraceful death to those who defame the country. |
| 2 | Nastaliq Urdu | خدا کرے سارے دانت ٹوٹ جائے اس کے مسکرا کر جب وہ کسی اور لڑکی کو دیکھے (ﷺﷺ) |
| | RU | Khuda kerey sarey daant toot jay us ky muskrat kr jab who kissi aur lerki ko dhekhey |
| | English translation | May God break all his teeth if he smiles when seeing a girl |

**Offensive**

| 1 | Nastaliq Urdu | اپنے فحش بے حیا بھائی سے پوچھ لیں۔ |
| | RU | Apney fehash bey haya bhai sey pooch len |
| | English translation | Ask your lewd brother. |
| 2 | Nastaliq Urdu | @MaryamNSharif چور کی بیٹی۔نس نس مین غریب قوم کاا خون بہ ربا |
| | RU | @MaryamNSharif choor ki beti. Nas nas mein Ghareeb qoom ka khoon beh reha |
| | English translation | @MaryamNSharif The daughter of a thief. The blood of the poor nation was flowing in the veins |

valuable insights into the contextual aspects that led to the assignment of these particular labels. This diverse representation not only enhances comprehension of the labeled dataset but also provides a detailed examination of the language nuances and cultural factors influencing the cyberbullying detection process.

## EXPERIMENTATION

This section provides a comprehensive overview of the conducted experiments aimed at evaluating and comparing the performance of both classical machine learning and deep learning techniques. The dataset was divided into training and testing sets and split ratio of 80% for training and 20% for testing is used. Cyberbullying detection approach of this

study consists of preprocessing, feature extraction and classification. The details of each step are given in the following sections.

## Preprocessing

Preprocessing is particularly important for tweets processing due to their unique characteristics, such as limited length, informal language, and specific conventions like hashtags and mentions. Therefore, before model construction raw data is cleaned up and transformed into a format that is appropriate for the following steps using a variety of approaches. Following preprocessing steps are applied.

Text cleaning: cleaning techniques are applied to standardize the textual data. This includes removing URLs, emails, and hashtags, as well as handling diacritics (if applicable). By removing irrelevant or noisy elements, the text becomes more manageable and consistent.

Tokenization: it involves breaking down the text into individual units, typically words or sub words. This process aids in creating a structured representation of the text, facilitating further analysis and feature extraction.

Stop words removal: common words are words that do not carry significant meaning in the context of the research study. Removing these words, such as articles, prepositions, and conjunctions, helps to reduce noise and improve the efficiency of subsequent analysis.

Lemmatization: Reducing words to their most basic or dictionary form is known as lemmatization. Words can be considered as a single entity by reverting them to their root forms, which allows for more precise analysis and model training.

## Feature extraction

The effectiveness of classification techniques relies on the choice of input feature set. This study employs three types of features: the classical bag of words (BoW) model, specifically term frequency inverse document frequency (TF-IDF), and state-of-the-art representations known as word embeddings. The BoW approach disregards grammar and identifies offensive sentences by checking for the presence of offensive, profane, threat, or abusive words. In this study, both unigram and bigram are explored as feature vectors for the bag of words model. Additionally, the significance of TF-IDF, a well-known information retrieval statistical technique, is investigated. The TF-IDF value of a word indicates its importance for classification within the *corpus*. For instance, cyberbullying annotated instances are expected to have a higher occurrence of offensive or abusive words. Word embedding is a dense vector representation method employed in deep learning models to capture word meanings based on contextual relationships. The embedding layer, which serves as the initial layer of the deep learning model, learns word representations from training data by considering adjacent words. When two words appear in similar contexts, their dense vector representations become more similar. Word embeddings effectively capture word semantics based on their contextual usage. In this study, pretrained Word2Vec embeddings are utilized to represent tweet-level features. We utilized 100-dimensional embeddings with a window size of 5, and the LSTM model

utilizes this embedding layer for word representation. For fastText word embedding of 300 dimension is utilized for unigram, bigram and trigrams.

## Classification techniques

In this section, we provide a comprehensive overview of both the traditional supervised machine learning techniques and the state-of-the-art deep learning techniques employed in this study for the detection of cyberbullying from tweets.

For traditional supervised machine learning methods, we utilized support vector machine (SVM), naïve Bayes (NB), and logistic regression. SVM is a popular algorithm that effectively separates instances of different classes by maximizing the margin between them. Naïve Bayes is a probabilistic classifier based on Bayes' theorem, assuming independence between features. Logistic regression is a widely-used linear model that estimates the probability of a certain class.

To leverage the power of deep learning in our cyberbullying detection task, we incorporated two state-of-the-art techniques: long short term memory (LSTM) and fastText. LSTM is a recurrent neural network (RNN) architecture that captures temporal dependencies in sequences, making it suitable for analyzing sequential data such as tweets. The LSTM cell equations for the input, output, and forget gates, as well as the cell state, are defined as follows:

$$i = \sigma\left(x_t U^i + s_{t-1} W^i\right)$$
$$f = \sigma\left(x_t U^f + s_{t-1} W^f\right)$$
$$o = \sigma(x_t U^o + s_{t-1} W^o)$$
$$g = \tanh(x_t U^g + s_{t-1} W^g)$$
$$c_t = c_{t-1} \circ f + g \circ i$$
$$s_t = \tanh(c_t) \circ o$$

where:

- $x_t$ is the input at time t,
- $h_t$ is the hidden state at time t,
- $c_t$ is the cell state at time t,
- $f_t$, $i_t$, $o_t$ are the forget, input, and output gates, respectively,
- $c_t$ is the new candidate value for the cell state,
- $\sigma$ is the sigmoid activation function.

LSTM can effectively model the context and capture long-term dependencies in the text. FastText, on the other hand, is a deep learning-based text classification technique that utilizes word embeddings and n-gram representations. The FastText algorithm is based on the bag of words (BoW) model and employs a linear classifier with a softmax function. The objective function for training FastText can be defined as:

$$J(\theta) = -\frac{1}{N} \sum_{i=1}^{N} \sum_{j=1}^{C} y_{ij} \log(p_{ij})$$

where:

- N is the number of training samples,
- C is the number of classes,
- $y_{ij}$ is a binary indicator of whether class
- j is the correct classification for sample i,
- $p_{ij}$ is the predicted probability of class j for sample i

It is known for its efficiency and ability to handle out-of-vocabulary words by utilizing subword information. FastText has gained popularity for its performance in various text classification tasks.

In this study for LSTM model, Adam optimizer with learning rate of 0.001 is used whereas learning rate = 0.5 is used in fastText model. LSTM model is trained for five epochs and 10 epochs are used for fastText training.

By employing a combination of traditional supervised machine learning methods and state-of-the-art deep learning techniques, we aim to explore the strengths and weaknesses of different approaches in the cyberbullying detection domain. This comprehensive analysis will enable us to gain insights into the effectiveness of these techniques and their suitability for handling the unique challenges posed by tweets in the context of cyberbullying detection.

## Evaluation metrics

The performance evaluation of the models is conducted using standard evaluation metrics, including recall (R), precision (P), and F1 score. These measures are employed to assess the models' ability to accurately detect cyberbullying in the Urdu tweets dataset. The equations for these metrics are defined as follows:

$$R = \frac{TP}{TP + FN}$$
$$P = \frac{TP}{TP + FP}$$
$$F1 = 2 \times \frac{P \times R}{P + R}$$

where:

- TP is the number of true positives,
- FP is the number of false positives, and
- FN is the number of false negatives.

These metrics provide a comprehensive evaluation, considering both the model's ability to correctly identify cyberbullying instances (precision) and its ability to capture all relevant instances (recall). The F1-score balances these two measures, offering a consolidated assessment of model performance.

## RESULTS AND DISCUSSION

In this section, results obtained from a series of experiments are presented. The goal of these experiments is to classify tweets into one of the seven categories: insult, offensive, name-calling, profane, threat, curse, or none. Table 6 presents the precision, recall and macro average F1 scores for the fine-grained multiclass classification. Upon examining the table, it is evident that NB performs the poorest in cyberbullying identification, achieving an F1 score of only 0.76. Conversely, fastText demonstrates the best performance, attaining F1 scores of approximately 0.842.

The analysis of the results presented in Table 6 reveals compelling evidence supporting the superiority of bigram bag of words (BOW) in achieving higher F1 scores compared to other feature sets, irrespective of the machine learning or deep learning model employed. The utilization of bigrams enables capturing contextual information, resulting in more accurate predictions of inappropriate or controversial content on social media platforms.

Furthermore, an intriguing observation emerges when considering precision, recall, and F1 scores across different data preprocessing techniques. The performance of the preprocessed text consistently outperforms that of the Raw-tweet data, indicating the value of carefully processing and cleaning the input text before applying the detection models.

This trend can be attributed to the effectiveness of text preprocessing techniques in removing noise, irrelevant information, and common language variations, thereby improving the model's ability to discern and classify instances of cyberbullying or offensive language accurately. The preprocessing steps, such as tokenization, stemming, or removing stop words, aid in standardizing the input text and enhancing the model's overall performance.

These findings highlight the importance of not only selecting appropriate feature sets but also emphasizing the significance of preprocessing strategies in achieving optimal results in cyberbullying detection tasks. By leveraging bigram BOW features and applying suitable preprocessing techniques, researchers and practitioners can enhance the accuracy and effectiveness of their detection models in identifying and addressing inappropriate or harmful content on social media platforms.

### Error analysis

In order to investigate the reasons behind misclassifications, an analysis of the incorrect predictions made by the best performing technique *i.e.*, fastText is conducted. This comprehensive analysis incorporates both quantitative and qualitative methodologies. The quantitative approach involves the use of a confusion matrix, which offers insights into the distribution of misclassified tweets, regardless of their content. In contrast, the qualitative approach delves deeper into the content of the misclassified tweets, enabling a more nuanced examination of the reasons behind the misclassifications. By combining these two approaches, a holistic understanding of the classification performance is achieved, encompassing both statistical patterns and the contextual information present in the misclassified tweets.

Table 7 provides an illustration of the findings, highlighting that the most significant confusion occurs between the "insult" and "none" classes in the context of cyberbullying

**Table 6 Cyberbullying detection in tweets.**

| Classifier | Features | Raw-text | | | Preprocessed-text | | |
|---|---|---|---|---|---|---|---|
| | | Recall | Precision | F1 | Recall | Precision | F1 |
| NB | Unigram BoW | 0.74 | 0.71 | 0.70 | 0.77 | 0.75 | 0.75 |
| | Bigram BoW | 0.75 | 0.72 | 0.72 | 0.78 | 0.77 | 0.76 |
| | TF-IDF | 0.69 | 0.74 | 0.57 | 0.70 | 0.75 | 0.59 |
| LR | Unigram BoW | 0.81 | 0.80 | 0.79 | 0.82 | 0.81 | 0.80 |
| | Bigram BoW | 0.81 | 0.81 | 0.79 | 0.83 | 0.82 | 0.81 |
| | TF-IDF | 0.75 | 0.76 | 0.70 | 0.77 | 0.77 | 0.73 |
| SVM | Unigram BoW | 0.81 | 0.81 | 0.81 | 0.82 | 0.82 | 0.82 |
| | Bigram BoW | 0.82 | 0.82 | 0.81 | 0.84 | 0.83 | 0.83 |
| | TF-IDF | 0.79 | 0.79 | 0.76 | 0.81 | 0.81 | 0.79 |
| LSTM | BoW | 0.74 | 0.80 | 0.77 | 0.76 | 0.82 | 0.79 |
| | Word2Vec | 0.72 | 0.77 | 0.74 | 0.76 | 0.81 | 0.79 |
| FastText | Unigram | 0.818 | 0.818 | 0.818 | 0.831 | 0.831 | 0.831 |
| | Bigram | 0.816 | 0.816 | 0.816 | 0.842 | 0.842 | 0.842 |
| | Trigram | 0.812 | 0.812 | 0.812 | 0.837 | 0.837 | 0.837 |

**Table 7 Confusion matrix of cyberbullying prediction using the fastText model.**

| | Insult | None | Offensive | Curse | Threat | Name-calling | Profane |
|---|---|---|---|---|---|---|---|
| Insult | 162 | 133 | 6 | 0 | 0 | 5 | 1 |
| None | 52 | 1,611 | 10 | 0 | 2 | 11 | 4 |
| Offensive | 31 | 51 | 104 | 1 | 0 | 3 | 1 |
| Curse | 0 | 5 | 0 | 9 | 1 | 0 | 0 |
| Threat | 1 | 7 | 2 | 0 | 7 | 1 | 0 |
| Name-calling | 6 | 20 | 8 | 1 | 0 | 113 | 0 |
| Profane | 0 | 21 | 2 | 0 | 0 | 4 | 53 |

prediction. This suggests that the fastText model faced challenges in accurately identifying instances associated with bullying behavior.

In the qualitative analysis, two word cloud figures provide insights into the misclassification of tweets. Figure 1 presents a word cloud of bullying (insult) tweets that were classified as None, whereas word of non-cyberbullying tweets classified as bullying (insult) is shown in Fig. 2.

An interesting observation from this analysis is the significant overlap of vocabulary among the misclassified tweets. Notably, words such as 'America', 'Bad', 'Laikan(But)', frequently appear in these misclassified tweets. It is evident that these words are commonly utilized across multiple contexts. This analysis underscores the importance of context and the challenges faced in accurately classifying tweets. The presence of shared

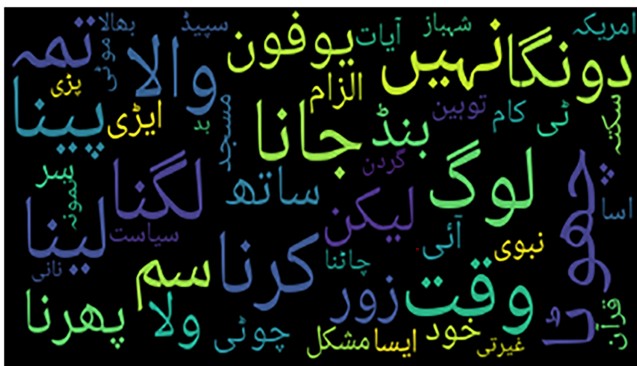

**Figure 1  Word cloud of Urdu tweets labeled as 'insulting' but classified as 'non-bullying'.**

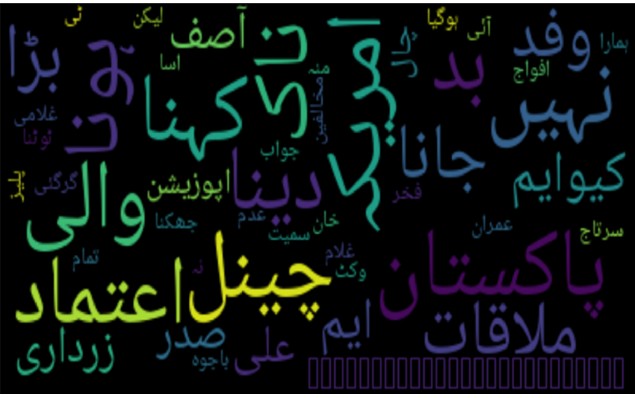

**Figure 2  Word cloud of Urdu non-bullying tweets classified as insulting.**

**Table 8  Comparative analysis with previous research work.**

| Study | Dataset size | Precision | Recall | F-measure |
|---|---|---|---|---|
| *Amjad et al. (2021)* | 3,564 | 66.50 | 75.25 | 70.56 |
| *Amjad et al. (2022)* | 3,500 | 79.47 | 79.37 | 79.42 |
| Proposed work | 12,428 | 84.2 | 84.2 | 84.2 |

vocabulary among misclassified tweets highlights the complexity involved in bullying detection tasks.

## Comparative analysis with previous research

In evaluating the efficacy of our proposed approach for threatening language detection within Urdu tweets, we conducted a comparative analysis with previous research works that specifically focused on this aspect. Notable studies, such as those conducted by *Amjad et al. (2021, 2022)*, have provided valuable insights into threat detection methodologies within the Urdu language.

Table 8 presents a comparative summary of key metrics, including precision, recall, and F-measure, specifically focusing on threatening language detection. Our proposed approach, with a dataset size of 12,428, achieves a precision, recall, and F-measure of 84.20%. Notably, this outperforms the results reported by *Amjad et al. (2021)* and demonstrates comparable performance to their 2022 study.

Our approach excels in the challenging task of threatening language detection, demonstrating superior precision, recall, and F-measure compared to previous studies.

# CONCLUSION

This research provides valuable insights into addressing the challenges of cyberbullying in the context of Urdu tweets. This study addresses a notable research gap by focusing on the detection of specific types of cyberbullying in the widely used Urdu language, which has received limited attention compared to the English language. Furthermore, the lack of benchmark labeled datasets in the literature for addressing cyberbullying in Urdu further emphasizes the significance of this study. This research makes significant contributions to the field of cybercrime detection on social media in two important ways. Firstly, this research generates the first publicly available cyberbullying Urdu *corpus*, which has been manually annotated based on iteratively developed guidelines. *Corpus* annotation guidelines can facilitate in future research to expand and enhance the *corpus*. Secondly, state-of-the-art machine learning and deep learning models are applied for the automatic classification of tweets. The collected dataset undergoes data cleaning and text preprocessing steps, and the experimental results demonstrate that the fastText model with bigram achieves the highest F1 score, indicating its effectiveness.

This study holds strong potential to inspire and motivate other researchers to address the pressing issue of cyberbullying detection in the Urdu language within the realm of social media. There are several promising directions for future research, such as enlarging the labeled dataset using guidelines and implementing data augmentation techniques. By doing so, a well-balanced dataset can be created, enabling a comprehensive evaluation of different techniques' effectiveness in combating cyberbullying specifically in the Urdu language. Another direction is to use advanced transformer-based models, such as GPT, RoBERTa, or XLNet. These models have shown promising results in various natural language processing tasks and can provide valuable insights into cyberbullying detection. By incorporating these recommendations into future research, a more comprehensive understanding of cyberbullying in Urdu tweets can be achieved, leading to improved model performance and enriched insights.

## Funding
The authors received no funding for this work.

## Competing Interests
The authors declare that they have no competing interests.

## Author Contributions

- Farah Adeeba conceived and designed the experiments, performed the experiments, prepared figures and/or tables, authored or reviewed drafts of the article, and approved the final draft.
- Muhammad Irfan Yousuf conceived and designed the experiments, authored or reviewed drafts of the article, and approved the final draft.
- Izza Anwer analyzed the data, authored or reviewed drafts of the article, and approved the final draft.
- Sardar Umair Tariq performed the experiments, performed the computation work, prepared figures and/or tables, authored or reviewed drafts of the article, and approved the final draft.
- Abdullah Ashfaq analyzed the data, authored or reviewed drafts of the article, and approved the final draft.
- Malik Naqeeb analyzed the data, authored or reviewed drafts of the article, and approved the final draft.

## Data Availability

The Urdu Cyber Bullying *Corpus* is available at GitHub and Zenodo:

- https://github.com/farahadeeba/cyberbullyingcorpus.
- Farah, A. (2024). Cyberbullying Data [Data set]. Zenodo. https://doi.org/10.5281/zenodo.10581022.

## Supplemental Information

Supplemental information for this article can be found online at http://dx.doi.org/10.7717/peerj-cs.1963#supplemental-information.

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
