# Peer review of "Addressing cyberbullying in Urdu tweets: a comprehensive dataset and detection system"

_PeerJ Computer Science, doi:10.7717/peerj-cs.1963_

## Round 0.1 · original submission · Major Revisions

Regarding the reviewers comments, especially reviewer 1, I recommend major revisions.

**Language Note:** PeerJ staff have identified that the English language needs to be improved. When you prepare your next revision, please either (i) have a colleague who is proficient in English and familiar with the subject matter review your manuscript, or (ii) contact a professional editing service to review your manuscript. PeerJ can provide language editing services - you can contact us at copyediting@peerj.com for pricing (be sure to provide your manuscript number and title). – PeerJ Staff

·

Basic reporting

### English language

Generally good, but still there are some refinements needed. For example:
> Internet
is misspelled in lowercase:
> internet

### Intro, background and references

From the abstract and introduction the reader gets the impression that machine learning and deep learning are two completely different approaches, while deep learning is a type of machine learning.

Introduction contains facts and statements that have to be proofed with literature and research resources. For example:
> Since the start of the Covid-19 lockdown, there has been a staggering 70% rise in the amount of bullying and hate speech among teens and children.

### Text structure

The structure of the text is too fragmented. There are also subsections that are too short, for example 3.4, 3.5.

## Figures and tables

Figures 3 and 4 are totally unclear and with no purpose. Figure 2 brings cares no information. Figure 1 is strange to be a flowchart since all actions are simply take one after another consecutively. Some tables contain wrongly encoded characters, maybe by the PDF converter.

### Raw data

Authors shared a Jupyter Notebook file with the experiment. This file contains no method, or algorithm.

Experimental design

### Originality of the research and scope of the journal

There is no originality in the proposed text. The problem is well known, and the machine learning methods, mentions by the authors are standard.

### Research questions definition

The text describes the execution on well-known existing methods on a standard natural language processing problem.

### Technical and ethical standards of the research

Technically, there is no single algorithm and even an mathematical construction in a text, dedicated to machine learning, which is not acceptable for a journal in computer science.

### Description detail sufficiency to replicate

The described experiments are trivial.

Validity of the findings

### Impact and novelty

No impact and novelty.

### Data provided robustness and statistical control

Data used in the research is statistically meaningful. There is no statistical analysis provided by the authors. Most of the information about the input data is contained in the tables.

### Conclusions

Conclusions are hard to be verified based on the text of the proposition.

Additional comments

Maybe this text can be addressed to a journal that is not focused on computer science.

Reviewer 2 ·

Basic reporting

In the related work section, authors should highlight on what grounds LSTM and FastText techniques are selected for experiments.

Experimental design

No comment

Validity of the findings

No comment

Additional comments

The authors have put sound efforts into addressing the research gap in Urdu cyberbullying detection, a low-resource language. The work can be accepted subject to the following modifications:


1. I suggest improving line 46 because these are objectives not the aims of the research and rewrite line 54 which is redundant. Lines 68-70 are redundant in the introduction section.
2. In the related work section, authors should highlight on what grounds LSTM and FastText techniques are selected for experiments.
3. The details of data augmentation techniques are missing from the data scraping section. They are generally discussed in line 153.
4. Figure 1 shows a block of “ Merge and shuffle”, why it is needed, an explanation is required in the data scraping section.
5. Although the results are compelling, the data analysis should be improved by adding a section that should compare your results with the previous research works (specified on lines 364-368 or any other relevant one).

---

## Round 0.2 · accepted · Accept

The comments from the Reviewer 1 were accurately checked. However, based on the information in the manuscript, the paper is Computer Science-related. It focuses on the development of a dataset and a detection system for cyberbullying in Urdu tweets, leveraging machine learning and deep learning techniques. This research aims to contribute to creating a safer digital environment by effectively detecting and classifying cyberbullying incidents on social media.

As the previous comments were addressed, the paper can be accepted.

·

Basic reporting

I am sorry, but i will stay with may previous opinion about this proposition. As the text is written, it is focused on the social and psychological impact of the problem. The few mathematical equations inserted by authors stay unrelated to the text, as if they are artificially placed. There is no computer science research in this text.

Experimental design

n/a

Validity of the findings

n/a

Additional comments

n/a

Reviewer 2 ·

Basic reporting

The introduction and related work sections of the article demonstrate sufficient knowledge of the field. The structure of the article is in an acceptable format. All the tables and figures are related to the content of the article.

Experimental design

Research questions are clearly defined. Experiments are rigorously performed and sufficient details are provided in the revised manuscript.

Validity of the findings

Results are validated by performing the comparison of the findings with the previous research works. The conclusions are appropriately stated, and connected to the original question investigated.